# Effectiveness of couple education and counseling on knowledge, attitude and uptake of cervical cancer screening service among women of child bearing age in Southern Ethiopia: A cluster randomized trial protocol

Samuel Yohannes Ayanto[1]*, Tefera Belachew[2], Muluemebet Abera Wordofa[1]

**1** Department of Population and Family Health, Jimma University, Jimma, Ethiopia, **2** Department of Nutrition and Dietetics, Jimma University, Jimma, Ethiopia

* contactsamijohn@gmail.com

## Abstract

### Background

Cervical cancer occurred nearly in 570 000 women and 311 000 women died from the disease worldwide in 2018. Of the new cases diagnosed globally in 2012, approximately 85% of the burden took place in low- and middle-income countries. Human Papilloma virus is the necessary cause for the development of cervical cancer and the majority of these infections resolves naturally but progress to precancerous lesions whenever there is persistence and delay in treatment. Majority of the cervical cancer cases, over 80% in sub-Saharan Africa including Ethiopia, have been detected at a late stage mainly due to poor early preventive measures. Therefore, utilization of early preventive measures could increase timely detection and treatment of precancerous changes and significantly reduce morbidity & mortality due to advanced disease.

### Methods

In this interventional study we will randomly assign 16 clusters (kebeles) in to the intervention and the control arm using block randomization. The study will employ a cluster randomized controlled trial. Women are eligible to participate in this study when they satisfy certain eligibility criteria; being in the age range of 30–49 years, no history of hysterectomy, did not receive cervical cancer or pre-cancer treatment and non-pregnant. Home based couple education and counseling will be provided to the eligible participants within the intervention group, while the control group receives standard of care. Base line and end line surveys will be completed by interviewing 288 eligible women to evaluate the effect of couple education and counseling on the knowledge, attitude and cervical cancer screening uptake. Generally the intervention lasts for six months. The results of baseline & end line surveys will be compared between the groups to determine the effectiveness of the intervention. Blinding is not possible due to the clustering of the trial arms.

**Data Availability Statement:** No datasets were generated or analyzed during the current study. All

relevant data from this study will be made available upon study completion.

**Funding:** This research will be carried out by financial support from Jimma University. The University has no role over the design, data collection, management and analysis, report writing, interpretation and the decision to submit the report for publication process.

**Competing interests:** The authors declare that they have no competing interests.

**Abbreviations:** FMOH, Federal Ministry of Health; HBM, Health Belief model; HDI, Human development Index; HIV, Human Immunodeficiency Virus; HPV: Human Papilloma Virus; LMICs, Low and Middle Income Countries; SSA: Sub Saharan Africa; TASH, Tikur Anbesa specialized hospital; VIA: Visual Inspection with Acetic acid.

## Discussion

Findings of the study will inform the regional or national scale up of the intervention modality to achieve the screening targets set by the Ethiopian government and world health organization.

## Trial registration

PACTR, PACTR202108529472385. Registered on 05 August 2021, https://pactr.samrc.ac.za/TrialDisplay.aspx?TrialID=16037

## Introduction

Human Papilloma Virus (HPV) is the necessary cause for the development of cervical cancer. Nearly all sexually active women are infected with HPV during their lifetime but majority of the infections resolve naturally within 24 months [1]. However, approximately 12% of these acute infections become persistent and can progress to precancerous lesions or invasive cervical cancer over decades, when not detected and treated early [2]. https://www.ncbi.nlm.nih.gov/pmc/articles/PMC5519520/pdf/fpubh-05-00178.pdf The precancerous lesions progress to advanced cancer stages usually within 10 to 20 years. This extended course in the progression of HPV infection to advanced cancerous stage provides an opportunity for the implementation of effective screening programs to prevent the development of cervical cancer through early detection and treatment modalities [1].

In 2012, the global cancer statistics showed that there were an estimated 527,600 new cervical cancer cases and 265,700 deaths worldwide [3]. Also, in 2018, cervical cancer occurred nearly in 570 000 women and 311 000 women died from the disease globally [4]. Cervical cancer in low- and middle-income countries (LMICs) accounted for approximately 85% of new cases and 87% of deaths that occurred globally in 2012 [5]. But in resource-rich countries the incidence and mortality due to cervical cancer were two to four times lower than what had been seen in resource scarce countries. The highest disease burden was demonstrated in southern and eastern Africa [4]. These statistics describe that there have been disproportionately heavy burden of cervical cancer among women in less developed regions of the world [1].

Larger proportions of cervical cancer cases and deaths occurred in Sub-Saharan African countries in 2018. Of the highest regional incidence and mortality rates observed in Africa, Eastern Africa shared the highest mortality rate due to cervical cancer in the year [6]. Also studies reported that the highest prevalence of cervical infection with HPV is recorded in Sub-Saharan Africa (SSA) countries [7]. Based on the projections made by World Health Organization (WHO), cervical cancer will be responsible for 443,000 deaths in 2030 globally [8] of which 98% will occur in low income countries, with SSA facing the highest number of deaths [5].

In Ethiopia, in a trend analysis of the cancer registry data, 5293 cervical cancer cases were diagnosed between 1997 and 2012 and accounted for 31.8% of all new cancer cases [9]. Also the trend analysis of cancer from 1998 to 2010 in Ethiopia showed that, malignancy involving cervix is among the leading malignancies in the country [10]. The estimated number of new cases of cervical cancer was 6047 with age specific incidence rate of 22% that accounts for about 20% of all identified female cancer cases in 2015 [11]. According to world's summary report, about 6,300 new cervical cancer cases and 4,884 deaths due to cervical cancer occur

each year in Ethiopia that makes cervical cancer the second-most common, and the second-most deadly cancer in the country among the target group [12].

Cervical cancer incidence and mortality have been considerably reduced in high resource countries during the last few decades. This is mainly due to the implementation of screening packages for the detection of precancerous cervical lesions and HPV infection. The availability of improved treatment options also played their role in this regard [13]. However, in low- and middle income countries where access to screening and treatment services is limited, cervical cancer remains a significant public health problem [6].

As a cervical cancer preventive measure, the Ethiopian Federal Ministry of Health (FMOH) in collaboration with the Pathfinder piloted Visual Inspection with Acetic acid (VIA) screening services combined with access to cryotherapy in 2009 for people living with Human Immunodeficiency Virus (HIV). These services have been later on scaled up into public healthcare facilities and standardized with the subsequent development of comprehensive cervical cancer prevention and control guideline in 2015 [14]. Nearly 1% of age-eligible women ever received screening in Ethiopia before the implementation of this guideline [15]. But more recent studies reported an uptake of 9.9% –15.5% in selected populations in Southern and Southwest Ethiopia [16–18]. Though the progress shows a favorable trend, it is far away from 80% target coverage nationally set for the 30–49 years target population by 2020 [19] which calls urgent public health interventions.

Evidence suggests that the availability and utilization of screening programs combined with effective treatment options lead to a significant reduction in the morbidity and mortality associated with advanced cervical cancer. In Ethiopia, despite the availability of screening services, only few of the eligible women underwent screening for cervical pre-cancer. Moreover studies have not been conducted to determine the effectiveness of locally adapted interventions which could increase cervical pre-cancer screening uptake among eligible women. Our research, therefore, aims to test the effectiveness of leaflet assisted home based couple education and counseling on knowledge, attitude and uptake of cervical cancer screening among eligible women.

## Methods

### Objectives

The objective of this study is to determine the effectiveness of leaflet assisted home based couple education and counseling in increasing the knowledge, attitude and uptake of cervical pre-cancer screening services among eligible women. Two groups of women are compared where the intervention group receives home based couple education and counseling while the control group receives the usual standard of care available in routine service delivery schemes.

### Trial design

This study is a two arm parallel cluster randomized trial that will be conducted in the Southern people regional state of Ethiopia.

### Study setting

The geographic location for our study is Kembata Tembaro and Hadiya zones which are located in the Southern Nations Nationalities and Peoples' Regional State. The total number of age eligible women (30–49 years) for cervical cancer screening in the study zones constitutes 19.2% of women of reproductive age group residing in the zones. Kembata Tembaro zone comprises 8 districts and 3 city administrations with the total number of 150 clusters or

kebeles, the smallest administrative units, of which 9 are located in the city administrations. The zone has one general hospital, four primary hospitals, 33 health centers and 138 health posts. Hadiya zone is administratively organized in to 13 districts and 4 city administrations having a total number of 329 clusters. The zone has one comprehensive specialized hospital, three primary hospitals, 61 health centers and 317 health posts.

The health care of women of child bearing age in general and maternal & child health care services in particular are of the main strategic pillars of the health service delivery efforts in the zones. Cervical cancer prevention and control activities are being integrated in to the routine health care delivery as an important health service package for the women's health. Currently, cervical cancer screening services are being implemented at selected hospitals and health centers of the zones where our research activities will base these facilities.

## Participants' eligibility

The study participants are women of child bearing age who are eligible for cervical cancer screening according to the Ethiopian national cervical cancer prevention and control guideline [14]. Accordingly women aged 30–49 years are targets for cervical cancer screening program and our research will be carried out within this program framework. Women will also satisfy the requirements of legal residency within their respective living quarters for at least six months, have not had received the screening services within the last 5 years, non-pregnant, beyond three months of postpartum, have not had hysterectomy, have not been diagnosed for any gynecological cancer including cervical cancer. These eligible women will be identified through censusing or health post records.

## Recruitment of participants

The Kembata Tembaro and Hadiya zones where the study districts have been situated belong to administrative structures in the Southern Nations, Nationalities and People Regional State of Ethiopia. These two zones were among the low performing zones as identified by the regional health bureau regarding utilization of cervical cancer screening services in 2021. This scenario drew the attention of the researchers to conduct this trial in these settings. Initially two districts, one from each, will be identified from the two zones for this study based on the availability of cervical cancer screening services. Then, sixteen geographically non-adjacent clusters or kebeles will be identified from the existing kebeles within the two study districts.

First, all possible non adjacent kebeles within each district will be listed. Then, among all non-adjacent kebeles, those which are relatively far apart to each other will be selected and included in the study. Therefore, non-adjacency and greater distance between kebeles will be the two selection criteria. Even though, it is difficult to indicate the distance in figures between included kebeles, we will leave at least one kebele between kebeles included for our research which may serve as buffer area between them. All the study clusters are kebeles located in the rural areas of the districts with the average population size of five thousand.

We will randomly assign the selected clusters in to intervention and control groups using block randomization technique as indicated in details below. We will conduct a census and/or use health post records to identify all the eligible women for cervical pre-cancer screening in the selected clusters. Subsequently, sampling frame will be created. Then, we will employ simple random sampling technique to select study participants from each cluster for each arm. Equal number of participants will be selected from each cluster.

Those women who consent to participate in the study will be included and requested to sign an informed consent to ensure voluntary participation. Selected women at each arm will be reached by health extension workers and the data collectors at their home. Once consent is

obtained, each participant will be interviewed to complete a baseline survey. The baseline questionnaire includes items on cervical cancer screening knowledge, attitudes, screening experiences, and socio-demographic variables. Similarly end line survey will be conducted at six months.

## Randomization

The clusters or kebeles are the units of randomization in our study. Initially, eight non-adjacent clusters from each district will be identified from the two districts. Before randomization, we will stratify the clusters by the study districts and create separate list of clusters alphabetically for each district. The stratification will be done to evenly distribute any known and unknown district level confounders across the study arms. Each cluster will be assigned a unique cluster code. Then, the statistician will assign the eight clusters in to two blocks of size 4 according to the order they appeared alphabetically. The statistician will randomly select the randomization sequence of clusters for each block using the sealed lots of the six possible permutations within the block.

The clusters in each block will be randomly assigned to intervention and control arms according to the randomization sequence evident by the selected permutation within in the block for the stratum. We will repeat the same process for blocks of clusters in the other stratum. Consequently, four clusters will be obtained from each district which will form eight clusters to the intervention arm and eight clusters to the control arm maintaining 1:1 random allocation ratio. Both the stratification and randomization techniques will help actual and potential confounding factors distribute evenly across the study arms which will eventually ensure comparability of the arms.

The statistician will be made unaware of the actual study arms to mask the knowledge about which group will receive the intervention and which group will receive the usual care. This will be achieved by representing the study arms and clusters by confidential codes. Health extension workers will carry out identification of eligible participants' for the study and subsequent intervention administration activities. Assessors will also be made blind to the nature of clusters with respect to intervention administration.

## Contamination reduction

When women from intervention kebeles join those women in the control kebeles during social occasions including funeral, wedding, marketing etc., there will be a possibility to contaminate the intervention messages through informal discussions of ideas regarding the intervention activities. As a result individuals from control kebeles may gain some intervention messages which may affect the intervention effect unfavorably. To reduce such an undesirable effect of information contamination, a buffer zone will be used to create non-adjacent clusters of the two study arms using clusters that will not take part in our study.

## Intervention description

The full version of the proposed intervention modality is brochure assisted home based education and counseling followed by formal invitation for cervical pre-cancer screening. The intervention will be implemented among research participants in the intervention arm. All the selected research participants who consented to participate in the intervention will be provided with the proposed intervention. Generally, the intervention will be delivered to the woman at three contact points during the intervention period. Of these contacts, the husband will take part during the second intervention schedule to receive education and counseling together with his wife. He with his wife will receive education on the key issues of cervical cancer and

its screening. The husband will also be counseled on the importance and the way how he will provide support and encouragement to his wife for cervical pre-cancer screening. Finally, the woman will receive educational material during the first visit and formal letter of invitation, during the last visit, which reinforces key messages of the intervention and importance of screening for cervical pre-cancer.

The educational and counseling material is organized to address susceptibility to cervical cancer, its seriousness, the benefits of screening and barriers to screening with the objective to build cues to action and develop woman's self-efficacy using health belief model. The intervention is organized and designed to bridge the knowledge gap and positively influence women's belief system related to cervical cancer and its screening. This, in effect would bring positive behavior change among women to use cervical pre-cancer screening services.

The educational brochure consists of five sections. Section one provides general information on the definitions, magnitude and incidence of the disease. Section two gives information and knowledge of risk factors, signs and symptoms, complications, and preventive modalities of the disease. Section three offers an explanation of the eligibility criteria, screening schedule, benefits and barriers to screening services to encourage participants to adopt positive behaviors. Section four explains the meanings of screening results, available treatment options, cost and location of the service. Finally, section five explains the importance of male involvement in woman's cervical pre-cancer screening uptake.

## Implementation of the intervention

Each woman will receive a total of three contacts during the intervention period. Accordingly, each eligible woman in the intervention clusters will be physically visited three times at her residential home. During the initial contact, a maximum of 45 minutes leaflet guided education & counseling session will be held with the woman to convey information on cervical cancer and its screening and counsel the woman to encourage the uptake of screening service. At the end of the session, discussion will be held with the woman to address any questions and concerns she may have. The woman will also be provided with the educational brochure for further reading at least once per week by themselves or to be read by any literate person within the family or neighborhood. The date of the subsequent follow up visit will be made consensually with the emphasis to attend the session together with her husband.

A follow up visit will be made to each woman one month after the initial visit with the objectives to re-emphasize important points of the material and make encouragement for screening. During this visit both the woman and her husband together will receive key information on cervical cancer and the importance of its screening. The husband will also be counseled on the importance and the way how the woman receives his support and encouragement to get screened for cervical cancer. Additionally any misconceptions related with the information will be corrected and barriers to screening will be addressed during this visit. Repeated follow up visits will be made in a case of absence of the couples.

Finally, the second follow up visit, which is the last visit of the proposed intervention modality, will be made one month after the first follow up visit to convey key messages about cervical cancer and its screening and address any unresolved concerns related to cervical cancer and its screening. Also, a formal letter of invitation will be granted during this visit for free screening services available in the nearby health facility. During all the visits eligible women will be asked key questions at the end of education and counseling session to check their comprehension. Also screening preparedness plan will be continuously negotiated with the woman at each visit as an encouragement for cervical cancer screening.

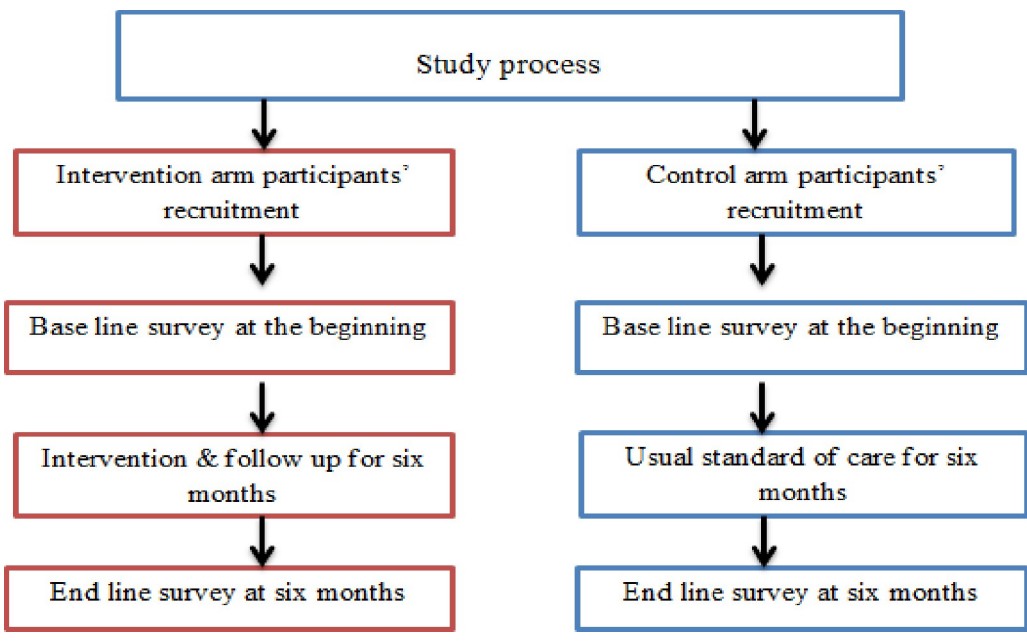

**Fig 1. Implementation framework of the study process.**

The intervention period lasts for six months in two phases. Phase one involves the first three months of active delivery of the proposed intervention at which time intervention will be provided to each woman in a monthly base. The second phase consists of the last three months of the intervention period. During the last three months women will not receive any intervention scheme but left unvisited with the objective to provide a period of rehearsal & translation of their knowledge in to practice. The health extension workers of the respective intervention clusters will be trained on the provision mechanisms of the intervention to couples at their homes. The overall framework of study implementation process has been depicted in the figure (Fig 1).

The nature of the health problem not being widely recognized among the rural communities convinced the research team to design three home visits in the trial package. Until recent years, in a majority of the cases cervical cancer has given less attention in the health system in our country and awareness level on cervical cancer among the rural communities is generally lower than expected. As a result we planned to test the effect of structured health education and counseling approach in a home environment with three home visits.

## Standard of care

Basically families in the study clusters receive a standard of care through home visits. Generally, the usual standards of care provided to families within the health extension program through home visits are preventive and pronmotive health care services. Health extension workers provide preventive and promotive services to households at home level with the objective to develop behavioral change among households on the health care services included in the health extension packages. These services are packaged under four major program areas such as disease prevention and control; family health; personal hygiene and environmental sanitation; and health education and communication [20].

Under disease prevention and control, for example, health extension workers promote behavioral change activities that target different health problems. They also promote the

utilization of health care services including disease screening services among the families. These services are common to all households within the kebele as the usual standard of care. Apart from these, the control group will not receive any innovative health care services such as structured home based couple education and counseling services during the study period. But the group will be exposed to the innovative intervention after the trial period through the routine services delivery schemes.

## Implementation of screening service

Those women who visit the screening facilities will receive the service according to the national guideline. Women will be provided with the specific information how they will access the service delivery location and whom they will contact within the health facility. Accordingly women who come from the study clusters for screening will be linked to the administrative procedures of the health facility to receive the screening service. A trained health professional will undergo assessment & counseling and conduct the screening procedure using visual inspection techniques according to the guideline. Any woman with positive precancerous screening results will receive treatment immediately after the results become available on the same day visit and be counseled to receive a follow up screening after one year. Those women who are tested negative will be counseled to receive a regular screening service after five years. All women who receive the screening will be registered on the format prepared for study purpose.

## Compliance parameter

Eligible women in the intervention clusters will be repeatedly visited in a case of absence to resolve the problem of compliance to full intervention package. Despite these efforts, due to different reasons, the eligible women in the intervention clusters may not fully comply with the proposed intervention as per the recommendations within the intervention package. This might exert undesirable effect on the uptake of the screening services. The impact of variability in the compliance to the proposed intervention requirements will be considered during analysis as a dose response function. Therefore, compliance checklist will be used to track the level of women's compliance to our proposed intervention package to account for during data analysis.

## Trial protocol summary

We have indicated a summarized version of the intervention description in the table (Table 1).

## Primary outcome

The primary outcome of this study is the completion of cervical cancer screening test within 6 months of the baseline assessment. Participants will be interviewed using the survey questionnaire and tracked via health records review for completing the screening service. The screening proportions will be determined for each study arm at the baseline & end line. The difference in

**Table 1. Summary of the intervention description.**

| Content of the intervention | Dosage | Frequency | Duration | Compliance parameter |
|---|---|---|---|---|
| Leaflet guided couple education & counseling | 30–45 minutes intervention | Every month | Two months | No of couples educated & counseled |
| Leaflet guided couple education & counseling and formal invitation | 30–45 minutes intervention | Once a month | One month | No of couples educated & counseled; and formally invited |

the screening proportions between intervention and control arms will be calculated. Finally, the effectiveness of the intervention will be measured as a difference of the differences in the screening uptake between the study arms.

## Secondary outcomes

We will also assess the progress of participants' knowledge and attitude by measuring their knowledge and attitude at baseline and end line. The effect of the intervention on participants' knowledge and attitude will be determined and compared between the study arms.

## Reliability of measures

Generally, the nature of the data collection tool we designed is adapted from different published studies and structured to fit the study context which contributes to the reliability of the measures. When we specifically come to the internal consistency of measurement items designed to measure attitude, its reliability will be assessed using Cronbach's Alpha and appropriate corrections will be applied based on the findings of the test. The cutoff value we will use for Cronbach's Alpha test is 0.70 or above. The other intervention we will employ to ensure the reliability of the measures will be training of the data collectors on the standards and techniques of data collection both before baseline and end line survey. Also, the same data collectors will assess study participants both during the base line and the end line survey.

## Participant timeline

The participant timeline suggesting enrolment, intervention delivery and assessment has been shown in the table (Table 2).

## Sample size

As a requirement, we identified the proportion of current cervical cancer screening service utilization from existing literatures to be 15.5% among age eligible women [18]. This has been taken as a baseline proportion for the study and considered to be the proportion for the control arm. We will expect an absolute difference of 20% increase in screening proportions between the study arms which gives the effect size or change in screening proportion due to the intervention to be 0.2. This produces the expected end line proportions of cervical cancer screening for the study to be 35.5% and considered as the proportion for the intervention arm.

The power of the study to detect the real statistical difference is set to be 80% and 5% significance level for one-tailed test has been used. We also considered the design effect of 2 to adjust for the loss of variability that would happen due to the effect of clustering and 5% as a compensation for incomplete and non-response rates that might happen during the data collection time. Considering the above assumptions and parameters, we arrived at a total of 288 study participants basically with the help of Gpower software. The required number of clusters we determined for the study is sixteen.

**Table 2. Participants' timeline of enrolment, intervention and assessment.**

| Activity | Month 1 | Month 2 | Month 3 | Month 4 | Month 5 | Month 6 | Month 7 | Month 8 |
|---|---|---|---|---|---|---|---|---|
| Participants Enrolment | ███ | | | | | | | |
| Baseline assessment | | ███ | | | | | | |
| Intervention | | ███ | ███ | ███ | ███ | ███ | ███ | ███ |
| End line Assessment | | | | | | | | ███ |

## Sampling procedure

Sixteen clusters (kebeles) will be selected from the two study districts where cervical cancer screening service is currently available. Sixteen clusters are taken based on the recommendation that taking fewer subjects from many clusters give better representation of the sample than taking many participants from fewer clusters. The clusters will be randomly assigned in to intervention and control groups. Census will be carried out in the selected Kebeles to identify women who are eligible for screening and create sampling frame. Then, simple random sampling technique will be performed to select study participants from each arm. We will select equal number of participants from each cluster which consequently leads to selection of 18 participants per cluster. The same sample of participants will be used at the end of the intervention phase, six months later, to measure the outcome variables (Fig 2).

## Data collection methods

Data will be collected using structured questionnaire designed for meeting the specific research purpose. The tool has been developed based on the research objectives from relevant literature sources. The data collection instrument will be pretested to check for its clarity, logical sequence, cultural appropriateness etc. and appropriate modification will be made accordingly. Participants will be interviewed to complete baseline survey after randomization of the clusters and end line survey at the end of the intervention period in six months. Participants will be measured for socio-demographic characteristics, knowledge, attitudes, and cervical cancer screening uptake.

Data will be collected through face to face interview technique using paper printed data collection tools contacting each woman at her residential home. Data collectors will go to each selected woman's home physically by carrying all the data collection tools and communicate verbally with the local language to get the required information. Two different teams will be assigned for the intervention administration and data collection for the purpose of masking

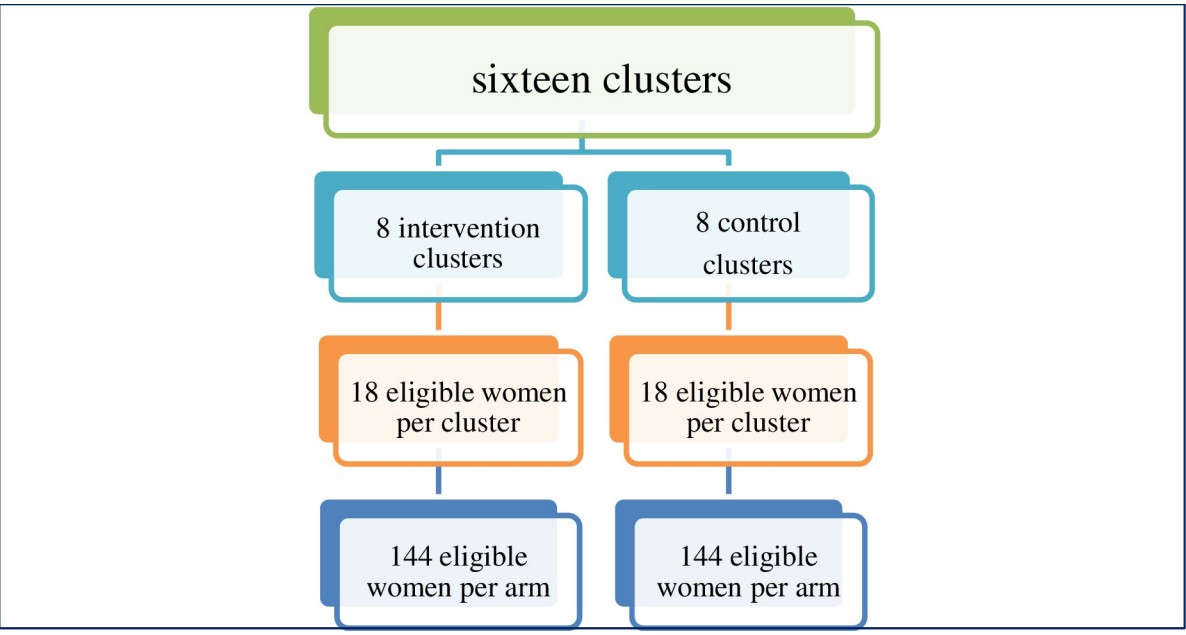

**Fig 2. Sampling procedures of the study participants.**

the intervention from data collectors. Both the baseline and end line data will be collected by the same individuals to ease the data gathering process.

## Confidentiality of the data

The information we will obtain from the participants will be kept confidential and used only for the purpose of the study. No personal identifiers will be recorded on the information and data recording sheet and only codes will be employed. The completed data checklists will be kept in a secure manner until it will be officially discarded.

## Data management

Specific cluster and individual codes will be assigned for each of the completed questionnaire. The data will be entered using Epi info version 7.2.4.0. The data will be edited and transported to either SPSS or STATA to carry out the desired statistical analysis.

## Statistical analysis

Base line screening proportions will be computed after randomization and before the intervention administration to measure the last 5 years screening performances of women in both arms with 95% confidence interval. The groups will be examined and compared at base line for any statistical differences in terms of base line participants' characteristics and their screening experiences. The baseline characteristics will be compared using chi-square test for categorical variables and the results will be indicated in chi-square value with p-value. The continuous variables of the groups at the baseline will be compared using independent sample t-tests and the results will be shown as mean, standard deviation and p-value. A one-sided P value of 0.05 will be used to determine statistical significance.

After completing the end line survey, proportions of women that will be screened during the trial period will be calculated with the corresponding confidence interval. We will also compare the before and after intervention scores of participants' knowledge and attitude of intervention group using paired sample t-test. A similar analysis will be done for the control group to compare the before and after intervention knowledge and attitude scores. Then we will employ the independent-sample t-test to determine the effectiveness of the intervention on knowledge and attitude by comparing participants' knowledge and attitude scores between the two groups. The results will be computed in terms of the value of the test statistic, means, standard deviations and p-value at 5%.

Finally, we will use Generalized Estimating Equation analysis technique to test the independent effect of the intervention on the uptake of cervical pre-cancer screening services. The effect size of the intervention, confidence interval and associated P-value will be determined during the analysis. Intention to treat approach will be employed to analyze the data.

## Data monitoring

Data will be monitored throughout the trial period with particular emphasis to baseline assessment, intervention period and end line assessment time. A team of field supervisors will carry out the responsibility of monitoring and auditing the overall trial data.

## Dissemination plan

The findings of this trial will be disseminated to relevant stake holders in the local community. The findings will also be communicated to the scientific community through publication of the results in the peer reviewed journals.

## Discussion

Cervical cancer screening is among the national maternal health intervention priorities in Ethiopia. Even with free screening services available, majority of eligible women do not use the service as expected. Effective locally adapted strategies to increase the uptake of cervical screening have not yet been tested across the nation. The development of nationally sound effective intervention strategies based on reliable research findings needs to be in place to increase the low coverage of the screening services evident currently. The results of this study could increase the uptake of the existing low cervical pre-cancer screening services utilization. Consequently, determining the effectiveness of leaflet assisted home based couple education and counseling on the uptake of cervical cancer screening with subsequent recommendations may help increase the rates of screening among the eligible group. This intervention strategy may also serve as a bridge in reducing cervical cancer morbidity & mortality among the risk groups.

## Supporting information

**S1 Checklist.**
(DOC)

**S1 Fig.**
(TIF)

**S1 Appendix.**
(DOCX)

**S1 Protocol.**
(DOCX)

## Acknowledgments

We would like to acknowledge Jimma University for granting ethical approval of this research activity.

## Author Contributions

**Conceptualization:** Samuel Yohannes Ayanto, Tefera Belachew, Muluemebet Abera Wordofa.

**Methodology:** Samuel Yohannes Ayanto, Tefera Belachew, Muluemebet Abera Wordofa.

**Writing – original draft:** Samuel Yohannes Ayanto.

**Writing – review & editing:** Tefera Belachew, Muluemebet Abera Wordofa.

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
