## [Decision Letter · Decision Letter 0]

24 Feb 2022

PONE-D-21-32347Effectiveness of Couple Education and Counseling on Knowledge, Attitude and Uptake of Cervical Cancer Screening Service among Women of Child Bearing Age in Southern Ethiopia: a cluster randomized trial protocolPLOS ONE

Dear Dr. Ayanto,

Thank you for submitting your manuscript to PLOS ONE. After careful consideration, we feel that it has merit but does not fully meet PLOS ONE’s publication criteria as it currently stands. Therefore, we invite you to submit a revised version of the manuscript that addresses the points raised during the review process.

We look forward to receiving your revised manuscript.

Kind regards,

Patricia Evelyn Fast, MD, Ph.D.

Academic Editor

PLOS ONE

Journal Requirements:

Additional Editor Comments (if provided):

Thank you for the opportunity to review this interesting approach. The reviewers have identified some ways in which you can clarify your presentation to help readers understand the protocol and judge its validity. Please pay particular attention to questions of how the kebeles are selected for intervention and how you will determine if they are basically comparable, as well as whether the sample size is adequate.

In addition, I would suggest that you be clear on the intervention to be tested. In the title and description this is described as couple counselling, however it appears that there is also a component of individual counselling that is much more intense than Standard of Care, with two home visits. Perhaps a description of 'standard of care' would help.

In addition, please state clearly what data will be available to readers--will all relevant data be included in the publication, or will some of it be deposited, in anonymized form, in a database or provided in some other way.

Some minor suggestions are appended as well.

Reviewers' comments:

Reviewer's Responses to Questions

**Comments to the Author**

1. Does the manuscript provide a valid rationale for the proposed study, with clearly identified and justified research questions?

Reviewer #1: Yes

Reviewer #2: Yes

Reviewer #3: Yes

2. Is the protocol technically sound and planned in a manner that will lead to a meaningful outcome and allow testing the stated hypotheses?

Reviewer #1: Yes

Reviewer #2: Yes

Reviewer #3: Yes

3. Is the methodology feasible and described in sufficient detail to allow the work to be replicable?

Reviewer #1: No

Reviewer #2: Yes

Reviewer #3: Yes

4. Have the authors described where all data underlying the findings will be made available when the study is complete?

Reviewer #1: No

Reviewer #2: Yes

Reviewer #3: Yes

5. Is the manuscript presented in an intelligible fashion and written in standard English?

Reviewer #1: Yes

Reviewer #2: Yes

Reviewer #3: Yes

6. Review Comments to the Author

You may also provide optional suggestions and comments to authors that they might find helpful in planning their study.

Reviewer #1: This is a good study and I look forward to seeing the results. A few suggestions to improve the paper:

1. In general, the tenses used makes it unclear whether the study has started or will be starting in future. Perhaps the authors could address the tenses in the paper to consistently be present or future tense and not both.

2. The Kebeles are the smallest administration unit - the authors can add more details to clarify what the Kebeles really are e.g. describe the hierarchy of the regions (i.e. province, district, zone, kebeles etc...) what the population is like for an average Kebele, average distance between Kebeles included, selection criteia for the Kebeles, were the Kebeles in the intervention and control arm matched on any characteristics prior to randomization? Is this a rural, urban or semi-urban setting

3. Recommend to add details of what the standards or control to the intervention is. This is to ensure that systems in place in the control arms are the same and no cluster has adopted an innovative way to improve uptake of screening for cancer services by women in a particular Kebele in the control arm.

4. Why did the authors choose Kembata Tembaro and Hadiya zone?

5. The sample size is not justified clearly. It doesn't seem that this sample size can lead to determination of effectiveness of intervention. Is it likely that the study results can lead in implementation of this strategy or a change in policy? Perhaps the authors should term this study as a pilot study i.e. "Pilot testing of education and counseling of knoweledge, attitude and uptake of cervical cancer screening..."

Reviewer #2: Thank you for tie opportunity to review this manuscript.

The protocol submitted for publication describes the procedures of a study of an important public health concern. cervical cancer is a significant cause of mortality among women in Africa. There are minor issues that need revision;

1. It would be important to collect some data from the male partners of the women in both arms and have at least qualitative data that can enrich the data that will arise from the assessments of the female volunteers.

2. Under data monitoring, is it better to consider calling it study monitoring as the monitoring doesn't only involve data monitoring but also monitoring other aspects such as ethical issues.

3 Other comments will be found in attachment

Reviewer #3: Only two issues to be addressed:

1. Reliability of the measures proposed to be used in the protocol should be documented in the manuscript (see its importance from a recent commentary: https://pubmed.ncbi.nlm.nih.gov/31253883/).

2. Statistical tests should be reported with more details (e.g., the effect size, confidence interval and statistical power, etc.)

7. PLOS authors have the option to publish the peer review history of their article (what does this mean?). If published, this will include your full peer review and any attached files.

Reviewer #1: **Yes: **Willam Kilembe

Reviewer #2: No

Reviewer #3: No

---

## [Author Response · Author response to Decision Letter 0]

30 Mar 2022

Responses to reviewers

Dear the editor and reviewers of our manuscript, we thank you very much for the commitment you have demonstrated and careful assessment of the manuscript for its potential maturity. We found all the specific comments and questions important for further improvement of the protocol and considered them in our manuscript. The way we considered them in our manuscript is described point by point in the following sections of this letter. Corrections have been made to the manuscript according to the specific comments. We also included responses to few questions in this rebuttal letter.

Question: Issues regarding references

Response: The entire reference list has been reviewed. There had been problem in three references listed in the 16th, 17th and 18th order. The problem we recognized is in the format of their citation. These references had been cited in Harvard style unlike the rest references which has been corrected to meet the standard of citation used in the manuscript. Reference number 12 has been replaced by most recent one. All the references have been edited and written in a complete manner.

Question: how the kebeles are selected for intervention and how you will determine if they are basically comparable, as well as whether the sample size is adequate

Response: We have provided additional elaborative descriptions to clarify these questions in the manuscript. For example, issues on how the intervention kebeles will be selected from the existing kebeles within the study districts and the issue of their comparability have been addressed. Concerning the sample size issue, we believe that the size of the sample is adequate for our study to make inferences on the bases of the study findings from the sample. With regard to this, we have tried to elaborate about the sample size we used in the methods section. As you read in the manuscript, we used appropriate statistical parameters based on the objectives of the study. For example, we used the effect size of 20% in which most researchers recommend for effectiveness studies as a rule of thumb. Also we used commonly recommended power of 80% and 95% confidence interval. In addition to maximize our sample size in order to overcome the effect of clustering, we considered the design effect of two to increase the sample size by 100%. Also non response rate of 5% have been added. Therefore, we addressed all the sample size concerns and requirements to maintain the minimum sample size required for our study to enable us infer the findings for the population.

Question: the intervention is much more intense than Standard of Care, with two home visits

Response: Basically families in the study clusters usually receive a standard of care through home visits. This include, mainly preventive and promotive health care services that target different health problems including cervical cancer. This is achieved through home visits made by health extension workers deployed at community level. Such education and counseling services in a usual care is not delivered using structured and organized educational material to guide the education & counseling service. The nature of the topic of interest is the main factor to design three visits in the trial package. For one thing, awareness level on cervical cancer in the rural setting is generally lower than expected. Until recent years, in a majority of the cases cervical cancer has given less attention in the health system in our country and not well recognized health problem among the rural community. As a result we want to test the effect of structured health education and counseling approach in a home environment with three home visits.

Question: please state clearly what data will be available to readers

Response: We described that data that will be generated from research participants and used for the research will be made available for the readers of our research work upon reasonable request. I.e. all relevant data that will be generated will be made ready for research purposes, publication requirements and other related matters upon request.

Question: In general, the tenses used makes it unclear whether the study has started or will be starting in future. Perhaps the authors could address the tenses in the paper to consistently be present or future tense and not both

Response: All the relevant tenses in the manuscript have been corrected to future tenses as to meet the requirements of our future plan of the study.

Question: the authors can add more details to clarify what the Kebeles really are e.g. describe the hierarchy of the regions (i.e. province, district, zone, kebeles etc...) what the population is like for an average Kebele, average distance between Kebeles included, selection criteia for the Kebeles, were the Kebeles in the intervention and control arm matched on any characteristics prior to randomization? Is this a rural, urban or semi-urban setting.

Response: all the concerns have been addressed within the manuscript in the methods section.

Question: Recommend to add details of what the standards or control to the intervention is. This is to ensure that systems in place in the control arms are the same and no cluster has adopted an innovative way to improve uptake of screening for cancer services by women in a particular Kebele in the control arm.

Response: These issues have been described in details in the methods section.

Question: Why did the authors choose Kembata Tembaro and Hadiya zone?

Response: These concerns have been justified in the methods section of the manuscript.

Question: The sample size is not justified clearly. It doesn't seem that this sample size can lead to determination of effectiveness of intervention. 

Responses: We tried to justify the sample size as much as we can in the sample size section. 

Question: Is it likely that the study results can lead in implementation of this strategy or a change in policy? Perhaps the authors should term this study as a pilot study i.e. "Pilot testing of education and counseling of knowledge, attitude and uptake of cervical cancer screening..."

Responses: This strategy or approach can be seen as an innovative type to be tested and will be scaled up in a larger context whenever effective in the trial. Basically, regardless of its label the study tests effectiveness of the proposed intervention. Labeling the study differently than the original makes no difference and does not change the nature and outcome of the study. Therefore, the authors preferred the labeling to be as it was in the original manuscript and approved protocol. Dear reviewer, does not this give meaning? This is not to disregard the labeling proposed by you but to indicate the preference to the original labeling by the research team. We apologize for the inconvenience.

Question: It would be important to collect some data from the male partners of the women in both arms and have at least qualitative data that can enrich the data that will arise from the assessments of the female volunteers.

Response: Basically, this is invaluable observation and we appreciate it but at this junction it will be practically beyond the scope of the study for which ethical review board has granted approval for its implementation. Male partners are included in this study only to create supportive environment to the female partner during the process of screening service utilization. In the intervention package, we are interested to see whether male involvement promotes cervical cancer screening utilization among women group. The trial is not intended to explore the opinions of the male partners. As a result we could not entertain the comment in the manuscript.

Question: Under data monitoring, is it better to consider calling it study monitoring as the monitoring doesn't only involve data monitoring but also monitoring other aspects such as ethical issues

Response: We recognize that your view may be important to improve the journal’s format. But we followed the journal’s guideline for the manuscript preparation. The journal’s format for the manuscript preparation dictates us to do so. Perhaps, the concept of data monitoring might apply particularly to the critical data management points in the research process. When actual research report is generated data monitoring applies to every aspect of data processing from data collection to report writing. It is within this understanding that we labeled the section as data monitoring.

Question: Other comments will be found in attachment 

Response: the comments provided within attached document were incorporated

Question: Reliability of the measures proposed to be used in the protocol should be documented in the manuscript

Response: this issue has been documented in the manuscript as per the comment under a separate heading known as “reliability of measures”.

Question: Statistical tests should be reported with more details (e.g., the effect size, confidence interval and statistical power, etc.)

Response: Statistical tests have been described in details to include important statistical parameters according to the comment indicated.

---

## [Decision Letter · Decision Letter 1]

15 Jun 2022

Effectiveness of Couple Education and Counseling on Knowledge, Attitude and Uptake of Cervical Cancer Screening Service among Women of Child Bearing Age in Southern Ethiopia: a cluster randomized trial protocol

PONE-D-21-32347R1

Dear Dr. Ayanto,

We’re pleased to inform you that your manuscript has been judged scientifically suitable for publication and will be formally accepted for publication once it meets all outstanding technical requirements.

Kind regards,

Patricia Evelyn Fast, MD, Ph.D.

Academic Editor

PLOS ONE

Additional Editor Comments (optional):

Thank you for your diligence in addressing the reviewers' comments. The paper is much improved in its clarity.

I have attached a few very minor wording comments (grammatical) for your consideration. Please also look carefully at the final comments of Reviewer # 2 for possible minor adjustments to the wording of the paper, to clarify the approach to literacy of participants.

'

Reviewers' comments:

Reviewer's Responses to Questions

**Comments to the Author**

1. Does the manuscript provide a valid rationale for the proposed study, with clearly identified and justified research questions?

Reviewer #1: Yes

Reviewer #2: Yes

2. Is the protocol technically sound and planned in a manner that will lead to a meaningful outcome and allow testing the stated hypotheses?

Reviewer #1: Yes

Reviewer #2: Yes

3. Is the methodology feasible and described in sufficient detail to allow the work to be replicable?

Reviewer #1: Yes

Reviewer #2: Yes

4. Have the authors described where all data underlying the findings will be made available when the study is complete?

Reviewer #1: Yes

Reviewer #2: No

5. Is the manuscript presented in an intelligible fashion and written in standard English?

Reviewer #1: Yes

Reviewer #2: Yes

6. Review Comments to the Author

You may also provide optional suggestions and comments to authors that they might find helpful in planning their study.

Reviewer #1: The authors have addressed the comments and questions raised and I have no further questions or comments.

Reviewer #2: Thank you for the chance for me to offer further review.

Minor comment.

We assume that the participants are all literate. If not, please ensure that the consents are translated and that there is provision for a witness on the consent form.

7. PLOS authors have the option to publish the peer review history of their article (what does this mean?). If published, this will include your full peer review and any attached files.

Reviewer #1: **Yes: **WILLIAM KILEMBE

Reviewer #2: No

---

## [Editor Report · Acceptance letter]

12 Jul 2022

PONE-D-21-32347R1 

Effectiveness of couple education and counseling on knowledge, attitude and uptake of cervical cancer screening service among women of child bearing age in Southern Ethiopia: a cluster randomized trial protocol 

Dear Dr. Ayanto:

I'm pleased to inform you that your manuscript has been deemed suitable for publication in PLOS ONE. Congratulations! Your manuscript is now with our production department. 

Kind regards, 

on behalf of

Dr. Patricia Evelyn Fast 

Academic Editor

PLOS ONE